# *Acinetobacter baumannii* Infections in Times of COVID-19 Pandemic

**DOI:** 10.3390/pathogens10081006

**Published:** 2021-08-10

**Authors:** Karyne Rangel, Thiago Pavoni Gomes Chagas, Salvatore Giovanni De-Simone

**Affiliations:** 1FIOCRUZ, Center for Technological Development in Health (CDTS), National Institute of Science and Technology for Innovation in Neglected Population Diseases (INCT-IDPN), Rio de Janeiro 21040-900, Brazil; 2Department of Pathology, Medical School, Federal Fluminense University, Niterói 24220-008, Brazil; thiago@id.uff.br; 3Department of Molecular and Cellular Biology, Biology Institute, Federal Fluminense University, Niterói 24220-008, Brazil

**Keywords:** *Acinetobacter baumannii*, infections, antimicrobial resistance, COVID-19

## Abstract

The COVID-19 pandemic has generated an overuse of antimicrobials in critically ill patients. *Acinetobacter baumannii* frequently causes nosocomial infections, particularly in intensive care units (ICUs), where the incidence has increased over time. Since the WHO declared the COVID-19 pandemic on 12 March 2020, the disease has spread rapidly, and many of the patients infected with SARS-CoV-2 needed to be admitted to the ICU. Bacterial co-pathogens are commonly identified in viral respiratory infections and are important causes of morbidity and mortality. However, we cannot neglect the increased incidence of antimicrobial resistance, which may be attributed to the excess use of antimicrobial agents during the COVID-19 pandemic. Patients with COVID-19 could be vulnerable to other infections owing to multiple comorbidities with severe COVID-19, prolonged hospitalization, and SARS-CoV-2-associated immune dysfunction. These patients have acquired secondary bacterial infections or superinfections, mainly bacteremia and urinary tract infections. This review will summarize the prevalence of *A. baumannii* coinfection and secondary infection in patients with COVID-19.

## 1. Introduction

After its recognition in Wuhan, China, in December 2019, coronavirus disease 2019 (COVID-19), caused by the severe acute respiratory syndrome coronavirus 2 (SARS-CoV-2), quickly spread across the globe. It became a global health threat [1,2,3]. As of 2 July 2021, SARS-CoV-2 had infected exceeding 180 million people and caused almost 4 million deaths around the world, according to the COVID-19 Weekly Epidemiological Update for that date [4]. Commonly, the clinical manifestations of COVID-19 are less severe, such as a mild upper respiratory tract disease or even asymptomatic infection [5,6]. However, SARS-CoV-2 infection can lead to severe and critical respiratory failure requiring mechanical ventilation, septic shock, or other organ dysfunction or loss that requires intensive care treatment [7]. Moreover, evidence suggests that coinfections and secondary infections can play an essential role in higher mortality risk during COVID-19 among a significant number of hospitalized patients [3,8,9].

Previous experiences with other coronaviruses and other respiratory pathogens have indicated this possibility of coinfection and secondary infections, especially by bacteria [10,11,12,13,14]. For example, *Streptococcus pneumoniae*, *Haemophilus influenzae*, and *Staphylococcus aureus* are the most commonly reported bacteria associated with co/secondary influenza infections [15,16]. In COVID-19 cases, while the prevalence of coinfections varied, the prevalence of secondary infections could be as high as 50% among non-surviving patients [2].

Hospitalization of COVID-19 patients, especially in intensive care units (ICUs), predisposes them to severe consequences such as healthcare-associated infections (HAIs) and/or secondary infections [17]. For example, COVID-19 patients admitted to ICUs with severe pulmonary symptoms can require the use of mechanical ventilation as part of supportive care. This use of mechanical ventilation can lead to ventilator-associated pneumonia (VAP), especially with multidrug-resistant bacteria such as *A. baumannii* [18]. *A. baumannii* infection cases have been reported in COVID-19 patients [19].

*A. baumannii* is an opportunistic pathogen mainly associated with HAI. Opportunistic pathogens can also cause superinfections, especially in combination with viral respiratory tract infections in hospitalized patients [19]. Belonging to the ESKAPE group (which includes *Enterococcus faecium*, *Staphylococcus aureus*, *Klebsiella pneumoniae*, *A. baumannii*, *Pseudomonas aeruginosa*, and *Enterobacter* spp.), *A. baumannii* stands out with its ability to effectively escape antibiotic treatments, affecting mainly immunocompromised and critically ill patients in ICUs [20]. Most commonly, *A. baumannii* infections manifest as VAP and bloodstream infections [21]. Given the clinical relevance and the antimicrobial resistance, the World Health Organization (WHO) appointed *A. baumannii* as a critical-priority pathogen that poses a significant threat to human health in 2017 [4]. Thus, we reviewed the medical literature to summarize and discuss the role played by *A. baumannii* coinfection and secondary infection in patients with COVID-19.

## 2. *Acinetobacter baumannii* Presentation

### 2.1. Characteristics of the Genus Acinetobacter

The genus *Acinetobacter* is a large and diverse group of biochemically, physiologically, and naturally multi-skilled bacteria. *Acinetobacter* spp., a ubiquitous coccobacillus genus, is characterized as glucose non-fermentative, non-motile, catalase-positive, oxidase-negative, and non-fastidious Gram-negative bacteria [22,23,24]. The taxonomy of this genus is complicated due to the numerous and closely related species, which are often impossible to distinguish from each other by phenotypic and chemotaxonomic methods [22,25]. Since the first description by Beijerinck in 1911 [25], this bacterial taxonomy has been reclassified under various names [21,26]. The *Acinetobacter* genus today contains 65 species with validly published names [27]. Currently, six species, namely, *A. calcoaceticus*, *A. baumannii*, *A. pittii*, *A. nosocomialis*, *A. seifertii*, and *A. lactucae* (a later heterotypic synonym of *A. dijkshoorniae*) [28,29] belonging to the Acb complex (*Acinetobacter calcoaceticus*–*Acinetobacter baumannii* complex) have been associated with human diseases [30]. Even though these species differ in their pathogenicity, antimicrobial resistance, and epidemiology [31], the Acb complex is physiologically and genetically highly related, making it difficult to distinguish them phenotypically with standard laboratory methods [32].

Highly found in the environment, bacteria of this genus can be recovered from different habitats, such as soil, surface water, foods, vegetables, and arthropods [33,34]. *Acinetobacter* can be retrieved as commensals of skin, wounds, and respiratory and gastrointestinal tracts [24,33,34]. In the hospital environment, *Acinetobacter* is also easily isolated, especially the *A. baumannii* species. When recovered from clinical samples, most species may have some significance as human pathogens [34].

Of all of the *Acinetobacter* species, *A. baumannii* is the most critical pathogenic member. *A. baumannii* has become a successful opportunistic pathogen and has emerged as a major cause of healthcare-associated infections (HAIs), mainly in critical and immunocompromised patients [23,35]. Furthermore, *A. pittii*, *A. nosocomialis*, *A. seifertii*, and *A. lactucae* are widespread and are occasionally associated with emerging important nosocomial pathogens involved in hospital-acquired infections [28,29,36].

### 2.2. Clinical Importance of A. baumannii

*A. baumannii* is mainly recognized as causing an extended range of HAIs, including pneumonia, bacteremia, urinary tract infections, wound infections, and meningitis [37]. A previous study estimated a global incidence of more than 1,000,000 cases of *A. baumannii* infections annually, of which 50% are carbapenem-resistant cases [38]. The National Healthcare Safety Network (NHSN) data for 2009–2010 indicate that *A. baumannii* was found in catheter-associated urinary tract infection, central line-associated bloodstream infection, surgical site infection, and ventilator-associated pneumonia [39]. These infections due to *A baumannii* may be associated with considerable mortality, varying from 8% to 35% [40]. Among these nosocomial infections, ventilator-associated pneumonia and bloodstream infections are the most important, with the highest mortality rates [41]. Risk factors associated with *A. baumannii* infections include prolonged hospitalization, intensive care unit admission, presence of devices or previous invasive procedures, prior use of antimicrobial agents, prior hospitalization, nursing home residence, older age, and prior colonization with *A. baumannii* [42,43,44,45,46,47].

According to previous studies, infected and colonized patients represent major reservoirs for the horizontal transmission and spread of *A. baumannii* in a hospital environment [48,49]. Furthermore, contaminated healthcare worker hands may also play a decisive role in this transmission. Thom et al. (2018) found that healthcare workers who provide care for patients known to be infected or colonized with *A. baumannii* exit the room with *A. baumannii* on their hands or gloves 30% of the time [50]. While infected and colonized patients are major reservoirs, *A baumannii* can be isolated from other hospital environmental sources. In the literature, different studies have been reported recovering *A. baumannii* isolates from various sources: sink and water taps [51], healthcare worker hand samples [52], hospital furniture, medical devices, and gloves [53].

According to previous studies, the ability of *A. baumannii* to persist and survive for long periods on surfaces and under dry conditions has made it a critical frequent cause of HAIs worldwide [54,55,56,57,58]. Moreover, outbreaks caused by *A. baumannii* isolates that displayed resistance to the vast majority of available antimicrobial agents have been increasingly reported during the last decades [26,59]. This ability to acquire antibiotic resistance determinants has propelled its clinical relevance [60].

While *A. baumannii* is an important nosocomial pathogen that can cause various diseases, community-acquired infections by this microorganism (including pneumonia and bacteremia) are less common, but it is associated with relatively high mortality [61]. The majority of community-acquired infection cases were reported in Australia and Asia: Australia [62,63,64], Hong Kong [65], India [66], Singapore [67], Korea [68], and Taiwan [69,70,71]. Risk factors for community-acquired infections caused by *A. baumannii* include chronic obstructive pulmonary disease, renal disease, diabetes mellitus, excessive alcohol consumption, and smoking [72,73]. Previous reports have also described the association of *A. baumannii* with infections consequent to war injuries and natural disaster victims [74,75,76].

### 2.3. Antimicrobial Resistance in A. baumannii

The drug-resistant *A. baumannii* and its susceptibility patterns have made empirical and therapeutic decisions even more complex [44]. High antimicrobial resistance rates are observed in Eastern and Southern Europe, Latin America, and many Asian countries [77]. This pathogen exhibits intrinsic resistance to different antibiotic classes, such as penicillins, macrolides, trimethoprim, and fosfomycin [78,79,80]. Moreover, *A. baumannii* displays a successful ability to acquire antimicrobial resistance [22]. Several studies have reported cephalosporin, carbapenem, aminoglycoside, and fluoroquinolone resistance in *A. baumannii* strains [78,81,82,83,84,85,86]. Mainly, carbapenem resistance in *A. baumannii* is an essential concern because this type of antimicrobial is the last line of defense used to treat infections caused by multidrug-resistant Gram-negative bacteria. Infections caused by carbapenem-resistant *Acinetobacter baumannii* (CRAb) cause more extended hospitalization, adverse outcomes, and increased costs than infections caused by carbapenem-susceptible strains [87,88,89].

Carbapenem resistance in *Acinetobacter* spp. is often associated with acquired carbapenemase production [81]. Among carbapenemases, Class D beta-lactamases—also called oxacillinases (OXAs)—are more frequent. Currently, the main groups of OXA-type carbapenemases identified in *A. baumannii* are OXA-23-like, OXA-24/40-like, OXA-58-like, OXA-143-like, and OXA-235-like groups, and the intrinsic chromosomal OXA-51-like group [90]. Acquired resistance to aminoglycosides (plasmid-borne aminoglycosides-modifying enzymes and 16S rRNA methylases) and fluoroquinolones (mutations in gyrA and/or parC) have been described in carbapenemase-producing *A. baumannii* strains [91]. Because of the increasing carbapenem resistance, second-line agents such as polymyxins and tigecycline have been considered for treating carbapenem-resistant *A. baumannii* infections [77]. However, resistance to polymyxins emerged and has been reported [92,93].

Antibiotic resistance mechanisms can be classified into three groups [94]: (1) Restriction or impediment of target access by reducing membrane permeability or increasing antibiotic efflux. (2) Direct antibiotic inactivation by a genetic mutation, post-translational modification, or enzymatic hydrolysis. (3) Access restriction through a structural change of membrane components or number of transmembrane segments.

The antibiotic resistance mechanisms of *A. baumannii* based on this classification are summarized in Figure 1.

## 3. SARS-CoV-2 and *A. baumannii*

### 3.1. Carbapenem-Resistant A. baumannii in Hospitals

Carbapenem-resistant *Acinetobacter baumannii* (CRAb), an opportunistic pathogen primarily associated with hospital-acquired infections, is an urgent public health threat [95]. CRAb readily contaminates the hospital environment and health care providers’ hands, can survive for prolonged periods on dry surfaces, and can be spread by asymptomatic colonization; these factors make CRAb outbreaks in acute care hospitals challenging to control [34,96,97,98]. It is resistant to common disinfectants, leading to outbreaks that are hard to contain and affect the most vulnerable and critically ill patients [34].

CRAb is the leading cause of morbi-mortality from infection in several European countries, with Italy being one of the top affected countries [99]. Compared to the pre-COVID-19 period, some authors observed a decreased antibiotic susceptibility in local pathogens during the first wave [100]. Others reported an increased risk of carbapenem-resistant infections in patients hospitalized with COVID-19 [18,99]. In both ICU and non-ICU settings, rigorous adherence to infection control practices is vital to discontinue the transmission of CRAb in hospitals [99,101,102,103]. Zhang et al. found that 55.6% of COVID-19 patients were coinfected with carbapenem-resistant *A. baumannii* (CRAb) in the ICU [104].

A recent study from a Mexican ICU with COVID-19 patients identified and characterized ESKAPE bacterias, detecting their possible clonal spread on medical devices, inert surfaces, medical personnel, and patients. *A. baumannii* was the most predominant ESKAPE member (52%) with all strains showing multidrug resistance (MDR). Moreover, the analysis of intergenic regions revealed a critical clonal distribution of *A. baumannii* (AdeABCRS+) in the ICU [105].

The outbreak of *A. baumannii* was described as one of the main determinants of severity and mortality of ICU patients in [106]. A New Jersey hospital experienced a large multidrug-resistant OXA-23 CRAb outbreak, primarily involving ICU patients, which extended across multiple units during a surge in COVID-19 cases [98]. A Japanese tertiary hospital reported an OXA-23-producing *A. baumannii* outbreak in 5 of 10 ICU beds, and all isolates had similar antibiotic susceptibility patterns with resistance to all β-lactams, including imipenem [107]. Of interest, an outbreak of *A. baumannii* was a determining factor in the increases of the incidence of infection and the morbi-mortality of ICU patients, with all strains multidrug resistant and only sensitive to colistin [108]. Gottesman et al. described a monoclonal outbreak of CRAb in two wards of a dedicated hospital to treat COVID-19 patients. All clinical (five cases) and environmental specimens (n = 24) belonged to international clonal lineage 2 (*bla*_OXA-66_ allele) and harbored the *bla*_OXA-24-like_ carbapenemase [109]. Shinohara et al. also report a monoclonal outbreak of an endemic strain of CRAb (14 cases) in a new COVID-19 ICU of a tertiary teaching hospital in southern Brazil [18]. Interestingly, CRAb outbreaks in COVID-19 patients seem to be effectively treated with a pre-optimized two-phage cocktail [110].

### 3.2. COVID-19 and A. baumannii Coinfections

The COVID-19 pandemic caused many immunocompromised individuals to be hospitalized, and some reports indicated that some COVID-19 patients were diagnosed with coinfections and secondary infections [93,94,111,112]. The incidence, prevalence, and characteristics of bacterial infection in these patients were not well understood and were raised as an important knowledge gap [113,114]. The specific nature and source of these infections have not yet been fully investigated; however, evidence indicates that multidrug-resistant bacteria are among the microbes responsible for developing these infections. The prevalence of coinfection was variable among COVID-19 patients in different studies. However, it could be up to 50% among non-survivors [2]. A systematic review and meta-analysis of bacterial coinfection and secondary infection in patients with COVID-19 reported 3.5% and 14.3% for coinfection and secondary infection, respectively. However, in general, bacterial infection was 6.9%, varying slightly in the patient population, ranging from 5.9% in hospitalized patients to 8.1% in critically ill patients [115]. Similarly, another study concluded that only 7% of hospitalized patients presented a bacterial coinfection with a high degree of heterogeneity, although this increased to 14% in studies that only included ICU patients [116,117].

Coinfection with *A. baumannii* secondary to SARS-CoV-2 infections has been reported multiple times in literature during the COVID-19 pandemic including Wuhan (China), France, Spain, Iran, Egypt, New York (USA), Italy, and Brazil (Table 1). The incidence of secondary infections (mostly lower respiratory tract infections) due to *A. baumannii* was said to be as high as 1% of hospitalized COVID-19 patients in an Italian hospital [118]. A descriptive study reported the exact incidence (1%) among hospitalized patients from Wuhan, China [119]. A simultaneous survey from Wuhan reported coinfection with *A. baumannii* in 1 out of 69 hospitalized patients (1.4%) with COVID-19 [120]. In addition, a recent study from a French ICU reported a 28% rate of bacterial coinfection with *A. baumannii* at 1 out of 92 (1.1%) in severe SARS-CoV-2 pneumonia patients [121]. A study conducted by Siyuan et al. (2021) investigated the frequency and characteristics of respiratory coinfections in COVID-19 patients in the ICU; they detected that *A. baumannii* and *S. aureus* were more frequently identified during late ICU admission [122].

Within the first few days of SARS-CoV-2 infection, critically ill patients with COVID-19 often develop pulmonary dysbiosis or respiratory tract distortion, which can progress further to a secondary bacterial or fungal infection a few weeks later [123,124,125]. A retrospective cohort study in a secondary care setting in Cambridge, UK, reported that a high percentage of patients with COVID-19 (9 out of 14) in the ICUs had confirmed secondary ventilator-associated pneumonia (VAP) [126]. Lescure et al. (2020) identified *A. baumannii* as the responsible agent in a COVID-19 patient of VAP [127]. A retrospective observational study of all patients admitted for COVID-19 in a University Hospital in Spain showed that 16% presented with fungal or bacterial coinfection/superinfection, and multidrug-resistant *A. baumannii* was the leading agent in respiratory infections and bacteremia with an outbreak contributing to this result [106]. Chen et al. (2020) also reported that bacterial and fungal coinfections in COVID-19 patients, including one patient that presented with an *A. baumannii* infection that was highly resistant to antibiotics, cause difficulties with anti-infective treatment, leading to the higher possibility of developing septic shock [119]. In a cohort study, the authors evaluated data from 212 severely ill COVID-19 patients admitted in a public tertiary hospital exclusively dedicated to attending COVID-19 patients during the pandemic and analyzed the association between fungal/bacterial coinfections and the mortality of these patients. *Acinetobacter* spp. was the second most isolated of the patients with positive bacterial cultures and was responsible for the third-highest mortality rate of COVID-19 patients suffering from these coinfections [108].

A study in an Iranian ICU reported coinfection with MDR *A. baumannii* in 17 out of 19 COVID-19 patients with high-level resistance to all antimicrobials tested, except colistin, which showed a resistance rate of 52%, although none of the patients survived [19]. Among 1495 COVID-19 patients hospitalized in Wuhan, 102 (6.8%) patients had acquired secondary bacterial infections, mainly due to *A. baumannii* (35.8%) with high resistance rates (91.2%), and almost half of them (49.0%, 50/102) died during hospitalization [8]. An Italian retrospective analysis of 32 COVID-19 ICU patients showed that 50% of patients developed an MDR infection during ICU stay. Overall, >80% of MDR bacteria isolated were Gram-negative bacilli, and the second most common pathogen isolated was carbapenem-resistant *A. baumannii* (CRAb). Four patients were coinfected with an extensively drug-resistant *A. baumannii* phenotype (12.5%) [128]. *A. baumannii* was detected in 20% of samples acquired from COVID-19 patients in an ICU in Beijing, China, during late ICU admission [122]. The highest incidence of MDR *A. baumannii* coinfection was documented in an Egyptian study (2.7%; 7 out of 260 patients hospitalized with COVID-19, susceptible only to tigecycline and fluoroquinolones, having resistance genes *NDM-1*, *TEM*, and *CTX-M*) [129]. Another study in hospitalized COVID-19 patients from Spain showed that coinfection with *A. baumannii* was apparent in 2.4% of hospitalized patients (17 out of 712; 16 of these 17 patients in ICU); it was the strongest determinant of mortality [106]. Cultrera et al. (2021) compared coinfections in critically ill patients with or without COVID-19, and *A. baumannii* was the most frequently isolated bacterium found in the three ICUs. In addition, CRAb infections in COVID-19-positive patients admitted to the ICU were more frequent compared to those that were COVID-19 negative [130]. The rapid expansion of the ICU to manage SARS-CoV-2 can potentially increase nosocomial infection rates in the hospital environment. As a result, there is great concern with bacterial coinfections in patients with COVID-19 as they significantly increase the morbidity and mortality of these patients. Thus, the early identification of bacterial infections will help to identify high-risk patients and determine the correct interventions to reduce mortality.

**Table 1 pathogens-10-01006-t001:** Incidence of co- and secondary infection of *A. baumannii* during COVID-19 pandemic reported in various countries.

Country/City	COVID-19 Patients	*A. baumannii* Coinfection n (%)	Other Pathogenic Organisms Found	Reference
China/Wuhan	102	57 (35.8%)	*K. pneumoniae* (30.8%), *Stenotrophomonas maltophilia* (6.3%) and others	[8]
Iran/Qom	90	17 (90%)	*S. aureus* (10%)	[19]
China/Wuhan	221	5 (55.6%)	*Escherichia coli*, *P. aeruginosa*, and *Enterococcus* (data not shown)	[103]
Spain/Valladolid	712	25 (18.7%)	*E. faecium* (17.2%) and others	[106]
Brazil/Minas Gerais	212	21 (32.8%)	*Staphylococcus* spp. (45.3%), *Pseudomonas* spp. (32.8%), *Stenotrophomonas* spp. (14.06%), *Klebsiella* spp. (12.5%), *Enterobacter* spp. (9.4%), *Enterococcus* spp. (9.4%), *E. coli* (6%).	[108]
France/Argenteuil	92	1 (3%)	*S. aureus* (31%), *Haemophilus influenzae* (22%), *Streptococcus pneumoniae* (19%), *Enterobacteriaceae* (16%), *P. aeruginosa* (6%), *Moraxella catarrhalis* (3%)	[121]
Egypt/Alrajhrt	260	28 (16.6%)	*S. aureus* (11.9%), *S. pneumoniae* (4.7%), *E. faecalis* (2.3%), *K. pneumoniae* (28.5%), *E. coli* (9.5%), *P. aeruginosa* (9.5%), *Enterobacter cloacae* (4.7%)	[129]
Italy/Milan	731	7 (30.4%)	*S. aureus* (69.7%), *E. coli* (21.7%)	[118]
China/Wuhan	99	1 (1%)	*K. pneumoniae* (1%), *Aspergillus flavus* (1%)	[119]
China/Wuhan	69	1 (1.4%)	*Candida albicans* (2.8%), *E. cloacae* (2.8%)	[120]
China/Beijing	20	10 (20%)	*Stenotrophomonas maltophilia* (28%), *P. aeruginosa* (28%)	[122]
France/Paris	5	1 (20%)	*A. flavus* (20%)	[127]
Italy/Naples	32	4 (19%)	*K. pneumoniae* (32%), *P. aeruginosa* (14%), *E. cloacae* (9%), *S. aureus* (4%), *E. faecium* (9%), *S. maltophilia* (9%), *E. faecalis* (4%)	[128]
Italy/Ferrara	28	17 (13.6%)	*E. faecalis* (14.2%), *E. faecium* (8%), *S. epidermidis* (33.6%), *S. maltophilia* (10.4%), *C. albicans* (23.2%)	[130]
Taiwan/Tainan	18	2 (11.1%)	*Streptococcus dysgalactiae* (11.1%), Influenza virus B (5.55%)	[131]

## 4. Conclusions

The worldwide outbreak of COVID-19 emerged in late 2019, and caused severe public health crises with considerably more significant morbidities and mortality rates, particularly in individuals with severe medical comorbidities and elderly populations. COVID-19 disease symptoms are diverse and may have varied manifestations among patients. In extreme cases, patients may develop pneumonia, acute respiratory distress syndrome, and multi-organ dysfunction. The screening and understanding of the proportion of COVID-19 patients with acute respiratory bacterial coinfection, especially VAP, is crucial for treating patients with COVID-19 and assisting in the responsible use of antibiotics, minimizing damaging consequences of overuse, and ensuring a better clinical outcome. In addition, this knowledge could have a significant impact on refining empirical antibiotic management guidelines for patients with COVID-19. The coinfection of SARS-CoV-2 and secondary infections with other microorganisms, such as *A. baumannii*, is an essential factor in COVID-19, and it can aggravate the progression and prognosis, especially in hospitalized patients. However, there are few reports about secondary infections and SARS-CoV-2 coinfections with bacteria; it is necessary to understand and strengthen the investigation of them. While there are many guidelines for diagnosing COVID-19 patients, bacterial coinfections and secondary infections have gaps and require further study.

This summarized review was designed to offer insight into *A. baumannii* coinfections and secondary infection in patients with COVID-19 using the most recent information available. In hospital settings, COVID-19 patients may face another health threat: antimicrobial-resistant bacteria such as *A. baumannii* and its carbapenem resistance pattern, which poses a considerable challenge to the health system. The isolation and the recognition of *A. baumannii* from COVID-19 patients highlights the importance of appropriate prevention and control practices. This microorganism potentially has important and severe implications for the clinical outcomes of these COVID-19 patients. Further studies are necessary to investigate the incidence, prevalence, and characteristics of COVID-19 coinfection and microbiological (especially *A. baumannii*) distribution.

## Figures and Tables

**Figure 1 pathogens-10-01006-f001:**
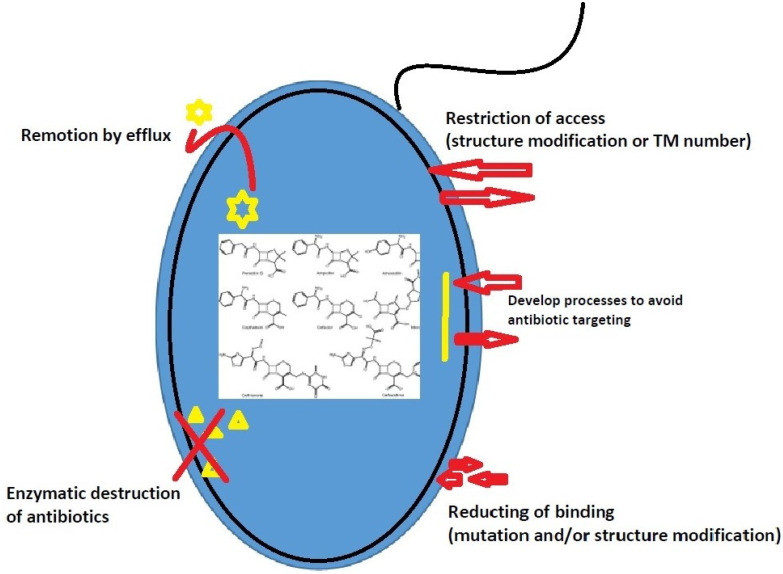
Schematic representation of the different mechanisms of resistance developed by *A. baumannii*. In the middle of the cell are the antibiotics.

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
