# Peer review of "Acinetobacter baumannii Infections in Times of COVID-19 Pandemic"

_pathogens, 2021, doi:10.3390/pathogens10081006_

Round 1
Reviewer 1 Report
The aim of this review paper is to summarize the prevalence of A. baumannii co-infections and secondary infections in COVID -19 patients.
The authors present adequate information on SARS-Cov-2 and its association in clinical practice with A. baumannii.
Some concerns:
1. This review highlights the role of carbapenem resistance (section 3.1). The choice is acknowledged as the increase in carbapenem resistance is an important concern because this type of antimicrobial is often considered the last line of defense.
However, what about adding information on resistance to other antibiotics such as fluoroquinolones, aminoglycosides and β-lactams? Since this is a review paper, adding relevant data will allow getting a global view of the problem.
2. There are some repetitions in the text. For example, there is no reason to repeat at the beginning of each section (3.1 and 3.2) the general principles about COVID19 outbreaks and co-infections with other microbes.
Try to delete these phrases to avoid repetition
Minor:
1. Please proofread to correct some grammatical errors / spelling of words.
Author Response
The aim of this review paper is to summarize the prevalence of A. baumannii co-infections and secondary infections in COVID -19 patients.
The authors present adequate information on SARS-Cov-2 and its association in clinical practice with A. baumannii.
Some concerns:
This review highlights the role of carbapenem resistance (section 3.1). The choice is acknowledged as the increase in carbapenem resistance is an important concern because this type of antimicrobial is often considered the last line of defense.
However, what about adding information on resistance to other antibiotics such as fluoroquinolones, aminoglycosides and β-lactams? Since this is a review paper, adding relevant data will allow getting a global view of the problem.
Thank you. All major changes in response to the refereers are written below and highlighted in the manuscript. Minor changes related to the use of English are not denoted. Some lines have changed because of changes suggested by refereers.
We agree with the referee and we include this information about resistance to other antibiotics in the section 2.3. and also introduce a figure with a schematic representation of the different resistance mechanisms developed by bacteria (Lines 141-174).
- There are some repetitions in the text. For example, there is no reason to repeat at the beginning of each section (3.1 and 3.2) the general principles about COVID19 outbreaks and co-infections with other microbes.
Try to delete these phrases to avoid repetition.
Thank you. This sentence has been removed:” Even the appearance of the SARS-CoV-2 pandemic, the world already required immediate and coordinated action to avoid the crisis of antimicrobial resistance and its related economic and health consequences. In particular” in the section 3.1.
Minor:
1. Please proofread to correct some grammatical errors / spelling of words.
Thank you. This was corrected.

Reviewer 2 Report
This manuscript reviews the relevance of Acinetobacter baumannii infections in the context of Covid-19 pandemia. Please correct grammatical or spelling errors marked in the attached document (marked in pink) and re-submit.

Author Response
This manuscript reviews the relevance of Acinetobacter baumannii infections in the context of Covid-19 pandemia. Please correct grammatical or spelling errors marked in the attached document (marked in pink) and re-submit.
Thank you. All major changes in response to the referees are written below and highlighted in the manuscript. Minor changes related to the use of English are not denoted. Some lines have changed because of changes suggested by referrers.
The grammatical and spelling errors marked in the attached document were corrected (Lines 36-37, 254).

Reviewer 3 Report
Rangel et al. present a review article describing current data on the prevalence of A. baumannii infections in COVID-19 patients and changing of antimicrobial resistance caused by overuse of antibiotics during pandemia. This topic is very hot and requires attention.
However, the presented review does not allow to get insights into this problem. I suggest rejecting the paper in this form and propose the authors to re-write it and include more relevant data regarding the changes of antibiotic resistance during COVID-19 pandemia and present the data in more structured way. I also suggest comparing the reports of A. baumannii prevalence in ICU units before (2018-2019) and during pandemia (2020-2021), since it is not easy to confirm that the changes were due to COVID-19 and not due to other factors.
Major comments:
- The review is very short and is not structured well. The chapters 2.1 and 2.2 describe the information that has already been reported in numerous papers and reviews. The clinical importance and high levels of resistance for A. baumannii are well known and could be mentioned in one or two paragraphs since these issues are not in the focus of the review.
- The chapter 3.2 contains the data for prevalence of A. baumannii in ICU patients with COVID-19. However, this chapter is simply a list of the works with percentage shown. No analysis is given and only a few reports confirmed the differences of co-infection rates between ICU patients with COVID and without COVID. It is very hard to follow this chapter – a table summarizing the reports would be better choice.
- Although this was mentioned in the abstract, no investigations were made regarding the change of A. baumannii antibiotic resistance during COVID-19 pandemia. Nothing was reported regarding the consequences of antibiotics overuse either. However, these issues are very important and are much more relevant than simple percentage values of the infections revealed.
- The conclusion is not detailed and most part of it is not connected to A. baumannii
Minor comments:
Title – “i” is missing in “infections”
p.1, line 29 – probably, a dot is missing before “SARS-Cov-2”
p.1, line 30 – should be “caused… deaths” or something – a verb is missing
p.1, line 42 – bacteria species should be defined fully (e.g., Streptococcus pneumoniae); short designations can be used in the later text
p.2., line 43 – please mention that the given bacterial species are associated with influenza (according to the references you provided), or add some references for other coronaviruses
- 2. line 44-45 – “most secondary infections could be as high as 50% among non-survivors patients” What does this mean? Has most infections been revealed in at least 50% of these patients, or at least one infection did? Please clarify
p.2., line 50 – “associated” after (V.A.P.) should be removed
p.4., line 131 – “pneumonia e bacteremia” – please correct
p.4, line 143 - “befre” – please correct
p.4, lines 156-157 – “However, others have reported an increased risk of carbapenem-resistant infections in patients hospitalized with COVID-19”. – please provide the references. Why did you put ‘however’ here? Decreased antibiotic susceptibility (from previous sentence) is not contradicted to increased risk of infection
p.4., lines 160-162 – the point is not clear. How many deaths were revealed in the first and the second groups? The differences between 5 out of 10 and 4 vs. 23 (as could be inferred from the presented data) are not significant, and the sample sizes are too small to make conclusions.
p.4, lines 165-168 – this sentence completely makes no sense. Please re-phrase and check grammar
Author Response
Rangel et al. present a review article describing current data on the prevalence of A. baumannii infections in COVID-19 patients and changing of antimicrobial resistance caused by overuse of antibiotics during pandemia. This topic is very hot and requires attention.
However, the presented review does not allow to get insights into this problem. I suggest rejecting the paper in this form and propose the authors to re-write it and include more relevant data regarding the changes of antibiotic resistance during COVID-19 pandemia and present the data in more structured way. I also suggest comparing the reports of A. baumannii prevalence in ICU units before (2018-2019) and during pandemia (2020-2021), since it is not easy to confirm that the changes were due to COVID-19 and not due to other factors.
Thank you. All major changes in response to the reviewer are written below and highlighted in the manuscript. Minor changes related to the use of English are not denoted. Some lines have changed because of changes suggested by refereers.
Thank you for your suggestion. Our main goal was to summarize the prevalence of A. baumannii co-infections and secondary infections in COVID -19 patients and we highlited the increased antimicrobial resistance due to the overuse of these drugs during the pandemic.
Major comments:
- The review is very short and is not structured well. The chapters 2.1 and 2.2 describe the information that has already been reported in numerous papers and reviews. The clinical importance and high levels of resistance for A. baumannii arewell known and could be mentioned in one or two paragraphs since these issues are not in the focus of the review.
Thank you. We agree with the referee and we include this paragraph in the review, section 2.3. and also introduce a figure with a schematic representation of the different resistance mechanisms developed by bacteria (Lines 141-174).
- The chapter 3.2 contains the data for prevalence of A. baumannii in ICU patients with COVID-19. However, this chapter is simply a list of the works with percentage shown. No analysis is given and only a few reports confirmed the differences of co-infection rates between ICU patients with COVID and without COVID. It is very hard to follow this chapter – a table summarizing the reports would be better choice.
Thank you. We inserted a table summarizing the reports (Line 237, 297).
- Although this was mentioned in the abstract, no investigations were made regarding the change of A. baumannii antibiotic resistance during COVID-19 pandemia. Nothing was reported regarding the consequences of antibiotics overuse either. However, these issues are very important and are much more relevant than simple percentage values of the infections revealed.
Several authors warned about a potential increase in multi-resistant microorganisms due to the high use of antimicrobials in patients with COVID-19 still in its early stages. The early and excessive use of antibiotics combined with the lack of rapid and accurate diagnostic resources creates the need for additional antibiotics if the patient presents clinical worsening, contributing to the use of broad-spectrum antimicrobials and possibly to an increase in bacterial resistance and its potential implications both in the hospital environment and in the community. However, we still do not have data to prove the change in antibiotic resistance of A. baumannii during the COVID-19 pandemic.
- The conclusion is not detailed and most part of it is not connected to A. baumannii.
Thank you. We revised the conclusion.
Minor comments:
Title – “i” is missing in “infections”
Thank you. This was corrected (Line 2).
p.1, line 29 – probably, a dot is missing before “SARS-Cov-2”
Thank you. This was corrected (Line 30).
p.1, line 30 – should be “caused… deaths” or something – a verb is missing
Thank you. This was corrected (Line 31).
p.1, line 42 – bacteria species should be defined fully (e.g., Streptococcus pneumoniae); short designations can be used in the later text
Thank you. This was corrected (Lines 42-43).
p.2., line 43 – please mention that the given bacterial species are associated with influenza (according to the references you provided), or add some references for other coronaviruses
Thank you. This was mentioned (Lines 44).
p.2. line 44-45 – “most secondary infections could be as high as 50% among non-survivors patients” What does this mean? Has most infections been revealed in at least 50% of these patients, or at least one infection did? Please clarify
According to authors, the secondary infection of died patients can reach 50%.
p.2., line 50 – “associated” after (V.A.P.) should be removed
Thank you. This was removed (Lines 52).
p.4., line 131 – “pneumonia e bacteremia” – please correct
Thank you. This was corrected (Lines 130-131).
p.4, line 143 - “befre” – please correct
Thank you. All this sentence was removed according to Referee 1.
p.4, lines 156-157 – “However, others have reported an increased risk of carbapenem-resistant infections in patients hospitalized with COVID-19”. – please provide the references. Why did you put ‘however’ here? Decreased antibiotic susceptibility (from the previous sentence) is not contradicted to increased risk of infection
Thank you. The word “however” was wrong and was removed (Line 188).
The references were provided (Line 189).
p.4., lines 160-162 – the point is not clear. How many deaths were revealed in the first and the second groups? The differences between 5 out of 10 and 4 vs. 23 (as could be inferred from the presented data) are not significant, and the sample sizes are too small to make conclusions.
Thank you. This sentence was rewrited (Line 191-193).
p.4, lines 165-168 – this sentence completely makes no sense. Please re-phrase and check grammar
Thank you. This sentence was rewrited (Line 194-199).

Round 2
Reviewer 1 Report
The authors have addressed my previous comments.
I have no further comments to make.
Author Response
Thank you, all corrections pointed by the referee 3 were considered in the text.
Reviewer 3 Report
My comments have been somehow addressed. I have several minor points yet.
p.1, line 31 – please change “277 834 caused deaths” to “caused 277 834 deaths”
p.1, line 35 – unclosed parenthesis “(“ – please remove or add another one
p.1, line 36 – “require intensive care manifestations” – makes no sense. Probably, “intensive care unit admission” or “intensive care treatment” is meant.
p.4., line 150 – not “multidrug Gram-negative bacteria”, but “multidrug-resistant Gram-negative bacteria”
Author Response
p.1, line 31 – please change “277 834 caused deaths” to “caused 277 834 deaths”
p.1, line 35 – unclosed parenthesis “(“ – please remove or add another one
p.1, line 36 – “require intensive care manifestations” – makes no sense. Probably, “intensive care unit admission” or “intensive care treatment” is meant.
p.4., line 150 – not “multidrug Gram-negative bacteria”, but “multidrug-resistant Gram-negative bacteria”
R: All corrections were made in the text, thank you.
